# Applications of Laser-Induced Fluorescence in Medicine

**DOI:** 10.3390/s22082956

**Published:** 2022-04-12

**Authors:** Mirosław Kwaśny, Aneta Bombalska

**Affiliations:** Institute of Optoelectronics, Military University of Technology, 00-908 Warsaw, Poland; miroslaw.kwasny@wat.edu.pl

**Keywords:** photodynamic therapy, fluorescence, laser, fluorophores, enamel

## Abstract

Fluorescence is the most sensitive spectroscopic method of analysis and fluorescence methods. However, classical analysis requires sampling. There are new needs for real-time analyses of biological materials, without the need for sampling. This article presents examples of proprietary applications of laser-induced fluorescence (LIF) in medicine with such methods. A classic example is the analysis of photosensitizers using the photodynamic treatment method (PDT). The level and kinetics of accumulation and excretion of sensitizers in the body are examined, as well as the optimal exposure time after the application of compounds. The LIF method is also used to analyze endogenous fluorophores; it has been used to detect neoplasms, e.g., lung cancer or gynecological and dermatological diseases. Furthermore, it is used for the diagnosis of early stages of tooth decay or detection of fungi. The article will present the construction of sensors based on the LIF method—fiber laser spectrometers and investigated fluorescence spectra in individual applications. Examples of fluorescence imaging, e.g., dermatological, and dental diagnostics and measuring systems will be presented. The advantage of the method is it has greater sensitivity and easily detects lesions early compared to the methods used in observing the material in reflected light.

## 1. Introduction

Fluorescence methods have played an important role in medicine and biochemistry for 50 years. DNA sequence analyses, immunofluorescence methods, flow cytometry, and analyses of vitamins, amino acids, porphyrins, pharmaceuticals and cations are among the classic examples of fluorescence technique applications.

The advantages of the method include its sensitivity, due to the intensity of fluorescence being proportional to the intensity of the excitation light, selectivity and ability to separate the emission spectra and excitation signals from the background. Another feature of modern methods is the possibility of using a variety of laser sources and optical fibers that transmit excitation and fluorescence radiation from anywhere in the human body or from the external environment.

The use of the LIF method for analyzing the state of biological tissues began in the 1990s. This method has been used for the diagnosis of skin diseases, atherosclerosis, kidney and urolithiasis and early stages of cancer [1,2]. “Optical biopsies”, as opposed to histopathological examinations, are non-invasive, do not require material sampling by fine-needle biopsy, the amount of analyzed material is unlimited, radiation is supplied and received via optical fibers, signals are measured in real time and the same areas can be analyzed repeatedly.

The mechanism of changes in the “autofluorescence” spectra of endogenous fluorophores is explained by their quantitative and qualitative differences in tissues, a change in their redox balance and depth of location, different content in tissues that absorb but do not fluoresce chromophores, changes in the extracellular matrix structure and the number of epithelial cell layers. Emission spectra of individual fluorophores in tissues are modified by the phenomena of light scattering and absorption of blood, which absorbs light in the visible part of the spectrum, and local changes in environmental parameters (pH, redox potential).

Quantitative laser-induced fluorescence (QLF) is a new diagnostic technique for enamel caries evaluation and the monitoring of mineral changes in initial caries [3]. The level of the fluorescence intensity of enamel changed in vivo is lower compared to healthy enamel. The decrease in fluorescence effect is mainly due to the scattering of excitation and emission light on damaged surfaces of hydroxyapatite. The coefficient of light scattering on decalcified enamel is 5–10 times greater than that on normal enamel. Research on the use of LIF has been ongoing for 40 years and was started by Bjelkhagen and Sundström [4]. They induced the fluorescence of a tooth material with nitrogen (N: 337 nm) and argon (Ar: 488, 514 nm) lasers. The intensity of the fluorescence emission in the blue–green range decreased with the increased degree of carious enamel decalcification for the excitation wavelengths given above. The field of this research is still continuing and is the subject of many current scientific articles [5,6,7,8,9,10,11].

In our research, a laser with a wavelength of 405 nm was used to excite the fluorescence of the enamel, and these studies showed a high correlation of the results with the measurements of mineral losses (R = 0.97).

The classic application of the LIF method is the quantitative assessment of the concentrations of photosensitizers in the photodiagnostics and photodynamic therapy methods.

The PDT method relies on the selective photooxidation of biological tissues by reactive oxygen species (ROS). A combination of an external photosensitizer, endogenous oxygen and red light produces singlet oxygen and free radicals, leading to necrosis and apoptosis of diseased cells [12,13]. The method of photodiagnostics is based on the localization of selectively absorbed PS in tissues using fluorescence methods.

Photodynamic therapy used with 5-aminolevulinic acid (ALA) or its methyl ester (MAL) is accepted worldwide for the treatment of skin cancers, non-cancerous diseases and photodiagnosis [14,15] This method has been implemented in urology, gynecology, neurosurgery, pulmonology and gastroenterology [16,17,18,19,20,21,22].

When applied exogenously, ALA, or its derivatives, is selectively metabolized to protoporphyrin IX (PpIX), a compound that gives a strong fluorescence, which serves as a basis for diagnosis [23]. The accumulation of PpIX in tissues occurs by avoiding the feedback control in the pathway of hem biosynthesis. Topical photodynamic therapy with ALA (ALA–PDT) reached approval status for actinic keratosis (AK) in US and Canada, whereas MAL–PDT is approved worldwide for AK, Bowen’s disease and Morbus Bowen in Europe and Australia.

Fluorescence methods are used to detect materials of biological origin. Our research indicates that the fluorescence of fungi also provides clinically useful information.

We presented methods for the quantification of sensitizer levels as well as examples of fluorescence imaging in in vivo studies of patients. The article presents the basics of the LIF method, the construction of apparatus, the characteristics of the organism’s endogenous fluorophores, examples of our own research on the use of the LIF method in medicine and further development directions.

## 2. Materials and Methods

### 2.1. Materials

Fungi samples (*Candida albicans*—ATTC 18804, *Aspergillus flavus*—ATTC 16883, *Penicillium chrysogenum*—ATTC 9179) were prepared as suspensions in water at the Military Institute of Hygiene and Epidemiology in Warsaw [24].

Photosensitizers—5-aminolevulinic acid hydrochloride (ALA) and amino acid derivatives protoporphyrin IX (PPIX)—were synthesized and purified at the Institute of Optoelectronics of the Military University of Technology (IOE MUT, Warsaw, Poland). Final preparations with a concentration of 10% ALA were prepared in the form of creams with the LIPOBAZA base. PPIX derivatives were used in the form of injection solutions in doses of 2.5 mg/kg body.

In vitro studies in dentistry were carried out using human teeth removed for various dental indications. Teeth with a completely preserved crown, without clinical caries changes and with carious spots were qualified for the study. The level of fluorescence was tested with an LESA 6 laser analyzer. Microradiography method was used to measure the depth of the lesion and loss of minerals. The teeth were sectioned for transverse microradiography. Tooth slices (250-thick) were sawn perpendicular to the enamel surface and then were manually ground to 70–80 thickness. Mineral content depth was measured with a microscope densitometer.

### 2.2. Measuring Apparatus

The analyzer was LESA5 spectrometer (BioSpec, Moscow, Russia) (Figure 1) [9], which was installed on the computer card, laser, fiber optic sensor (catheter), optical filters. Depending on the application, the following lasers were used: He-Ne (λ = 632 nm, 25 mW, BioSpec), II harmonic Nd: YAG (λ = 532 nm, 10 mW, BioSpec), semiconductor lasers (λ = 375 nm, 15 mW and λ = 405 nm, 25 mW, Power Technology, Little Rock, AR, USA), He-Cd (λ= 442 nm, 100 mW, Omnichrom, Rochester, NY, USA).

Fluorescence imaging system (Figure 2) consists of the following main components: CCD camera GP-KS162 (Panasonic, Osaka, Japan), xenon lamp with liquid fiber 300 W (Lasar, Warsaw), endoscope (Storc) and optical filters (IOE MUT).

The LIF method was used for the diagnosis of pathological changes in in vivo conditions on patients in Polish clinics that had approvals of the relevant bioethical committee.

## 3. Results

### 3.1. Spectral Characteristics of Fluorophores

Figure 3 shows the collective absorption and emission characteristics of the most important fluorophores (fluorescent chromophores) found in biological systems [25]. The tryptophan bands (components of elastin, collagen), FAD and NADH coenzymes and endogenous porphyrins can be clearly distinguished from the field of fluorescence excitation.

The sources of endogenous fluorescence in cells and biological tissues are aromatic amino acids, which are used to build proteins and coenzymes. Among the 20 amino acids from which proteins are built, only tryptophan (TRP), tyrosine (TYR) and phenylalanine (PHE) have fluorescence in the UV region 1.

The main component of bone, hydroxyapatite, has strong fluorescence properties in hard tissues. In many disease cases, increased levels of metalloporphyrins are observed [26].

The coenzymes FAD and FMN and vitamin B2 absorb light with a wavelength of around 450 nm and emission at a wavelength of around 530 nm. Unlike NADH, only the oxidized form of FAD shows fluorescence.

The phosphoryl derivative of vitamin B6 is another fluorescent coenzyme [27]. Vitamin B6 occurs in three forms with the same biological activity as pyridoxine, pyridoxal and pyridoxamine. Biologically active forms are phosphate derivatives of pyridoxamine and pyridoxal, which interact with enzymes active mainly in the transformation of amino acids (including racemization of optically active amino acids, transamination, decarboxylation, tryptophan synthesis).

An important group of fluorophores are pteridine derivatives, heterocyclic compounds containing several substituents in the basic pterin structure [28]. The pterin is composed of conjugated pyrazine and pyrimidine rings that contain carbonyl oxygen and an amino group. The pteridine system is widespread in nature because its derivatives are the basis for the coloration of the wings and eyes of insects, as well as the skin of amphibians and fish [25]. Folic acid, necessary to produce red blood cells above the bone marrow, is made up of the pteroyl group, p-aminobenzoic acid and glutamic acid. The pteridine system is found in bacteria and fungi.

### 3.2. Application of LIF in Dental Diagnostics

Typical single fluorescence characteristics consist of an excitation (equivalent to the absorption spectrum) and emission spectra. By changing the wavelength of the excitation radiation in the entire absorption range, an emission–excitation (EM–EX) matrix is obtained. It is the real spectral imprint of the tested sample. The method is of particular interest for the analysis of substances containing various fluorophores. In addition, it enables the selection of appropriate wavelengths for testing. Figure 4 shows the enamel and dentin EM–EX matrices.

The strongest fluorescence of enamel is obtained after excitation with radiation in the range close to UVB and violet.

The LIF spectra with selected excitation wavelengths are shown in Figure 5. The level of fluorescence of the enamel with caries is lower compared to the unchanged enamel. In the case of dental caries, an increased level of fluorescence is observed, which is associated with porphyrin derivatives generated by bacteria.

Figure 6 shows the relationship between the decrease in fluorescence intensity and other parameters characterizing the degree of caries: the depth of changes and the degree of mineral loss. Measurements of these parameters were carried out using the microradiography method, and the average decrease in the fluorescence of the demineralized area was determined using a LIF spectrometer with radiation excitation and a wavelength of 407 nm.

### 3.3. Clinical LIF Applications Using Endogenous Fluorophores

The LIF method based on the study of endogenous fluorophores is of greatest importance in pulmonology and dermatology. Neoplasms are characterized by lower fluorescence in the green range (about 530 nm) and a higher ratio of fluorescence in the red and green bands compared to healthy tissues. Lowering the level of autofluorescence in neoplastic tissues in the area of FAD emission is related, among other factors, to a greater metabolism of these tissues (an increase in the level of NADH and a decrease in the amount of the oxidized form of FAD). Figure 7 shows the decrease in tissue fluorescence in the case of pleural mesothelioma. LIF studies were conducted on 23 cases of lesions. The data were analyzed by performing ANOVA test comparisons between normal and tumor tissues (significance level α = 0.05.). There was statistically significant difference (*p* < 0.01) between these groups of tissues.

An interesting problem is the presence of increased levels of porphyrins in many diseases. This is evident in the case of porphyria. Increased accumulation of porphyrins in neoplastic tissues has been observed many times by the authors of this work in many skin diseases (senilis keratosis) or in advanced cervical neoplasms. The causes of the fluorescence tissue of the squamous cell carcinoma of the oral cavity are metalloporphyrins contained in bacteria (*Pseudomonas bacteria*).

Examples of the presence of elevated porphyrin levels are shown in Figure 8.

The conventional diagnosis of oral candidiasis is generally based on biopsy tissue; however, this technique is time-consuming. *Candida* is a pathogenic organism that may cause oral candidiasis upon disruption of the balance of flora. The disease is most commonly caused by an overgrowth of *Candida albicans* in the mouth [29]. Figure 9 shows the fluorescence characteristics of selected fungi.

### 3.4. Measurements of Photosensitizers in the PDD/PDT Method

The classic and most important application of the real-time fluorescence method is the analysis of photosensitizers in the photodiagnostics and photodynamic therapy methods. These studies include (i) localization and determination of the level of PS concentrations, (ii) kinetics of their accumulation and excretion over time, (iii) determination of the optimal time of therapeutic irradiation from the moment of introducing compounds into the body, (iv) photochemical distribution of sensitizers and (v) selection of therapeutic irradiation parameters.

An example of the kinetics of PPIX accumulation after ALA application in the case of a change in actinic keratosis is shown in Figure 10. These studies are necessary in respect of introducing new methods to the market to form ALA [30].

A sufficient level of PPIX for further therapeutic irradiation is obtained at least 2 h after the application of ALA. The topical introduction of an allergic to superficial dermatological changes is the easiest way. In the case of tumors of internal organs or lesions of greater thickness, it is necessary to inject photosensitizers. In cases where it is necessary to analyze changes in tissues of greater thickness, the use of a red laser is a better choice for fluorescence excitation due to greater light penetration.

Figure 11 shows an example of the use of the He-Ne laser for the photodiagnostics of cancer (Merkel carcinoma) 48 h after injecting the amino acid PPIX derivatives (2 mg/kg body mass) into the blood.

The PPD/PDT method has found application in gynecology. Figure 12 shows examples of the use of the LIF method in the treatment of vaginal and cervical lesions.

The fluorescence images of these changes are shown in Figure 13.

Apart from the research on the kinetics of photosensitizers’ accumulation in tissues, the LIF method is helpful in determining the light power density in the PDT method. During irradiation, the photochemical decomposition of porphyrins takes place, and this process depends on the intensity of the light. The photobleaching effect of the sensitizer as a function of irradiation is shown in Figure 14. When using a power density of 100 mW/cm^2^, the degradation of the sensitizer occurs much faster than at 40 mW/cm^2^, the therapeutic effect is insufficient and the treatment procedure must be repeated.

## 4. Discussion

Endogenous fluorophores that occur in the body and are the basis of autofluorescence can be divided into three groups based on the spectral range. Absorption in the UVB range (280–325 nm) is demonstrated by amino acids and proteins. Fluorometric analyses of these substances play an important role in biochemistry. In the LIF method, lasers such as He-Cd (325 nm), Nd YAG (266 nm) or tunable OPO or titanium [31] can be used to excite the fluorescence. The systems built are used to detect biological agents in the air, which are mainly used in military technology. The UVA (325–380 nm) and blue light ranges include fluorophores that contribute to the metabolism of the organism (NADH, FAD), pterins and porphyrins. For example, the ratio of NADH to FAD fluorescence is an indicator of the metabolic rate. For medical applications, the visible range is the most important. Violet or blue light excitation on tumor tissues in comparison to healthy ones shows a decreased level of fluorescence (Figure 7).

Hydroxyapatite, the tooth component, fluoresces within a wide spectral range, from 350 to 450 nm. Quantitative QLF methods have already found practical application, and imaging systems are already being built (e.g., Inspektor Research system, Bussum, The Netherlands) [32]. LIF spectrometers allow for more accurate analysis, are many times cheaper and allow every part of the mouth to be reached with optical fibers.

Caries is a complex pathological process, which entail the gradual loss of minerals from the hard tissues of the tooth. Under the conditions of ionic equilibrium, normal enamel undergoes continuous de- and remineralization processes, which do not cause changes in the enamel structure. If the pH drop in the oral cavity is not balanced, it causes disturbances in the biochemical balance and initiation of the destructive process under the influence of acid metabolites. The clinical symptom of early carious lesions is the appearance of whitish, opalescent spots on the enamel surface. However, the diagnostic effectiveness of most of the clinical methods used so far is unsatisfactory. Modern methods of radiological diagnostics currently available in clinical practice do not allow for the detection of changes related to the very early phase of enamel demineralization. The methods with high expectations include fluorescence induced by lasers.

The level of autofluorescence of healthy enamel in comparison to the carious enamel in vivo is higher when excited with lasers with wavelengths of 405 and 442 nm (Figure 5) In imaging systems, the decrease in the fluorescence of the carious area is visible as a dark contrast against the bright background of healthy enamel, which greatly facilitates the diagnosis. One of the important achievements is showing the influence of the depth of the lesion and loss of minerals in enamel on the decrease in fluorescence intensity (Figure 6). A measurable decrease in the fluorescence intensity is already visible at a 5% loss of enamel mineral. In such cases, it is possible to effectively remineralize the enamel with appropriate dental pastes, without the need for drilling. Very early caries diagnosis is the main advantage of the LIF method.

LESA is a PC-based spectroscopy system consisting of a laser source for fluorescence excitation, a miniature monochromator, multichannel CCD detector, an optic fiber sensor and a computer for data acquisition and processing (Figure 1). The entire fluorescence spectra range is recorded simultaneously. The intensity of fluorescence depends on the irradiance, the distance between the sensor and the light, and the position of the sensor. Thus, it is important to normalize the signals. The monochromator receives fluorescence emission and is scattered on the tissue by laser radiation, which is attenuated 10^3^ times by an appropriate optical filter. The monochromator receives fluorescence radiation and laser radiation scattered on the tissue, which is attenuated 10^3^ times with an appropriate optical filter. The obtained spectra are normalized to the laser signal. It is a convenient reference signal for the fluorescence measurements. The laser emission to fluorescence area ratio depends only on the concentration of the fluorophore (e.g., Figure 8 and Figure 9). Our goal was to develop a technique for quantifying fluorophores in all possible medical cases—for caries testing, PDT sensitizers and cancer by autofluorescence (endogenous sensitizers). The only such commercially available system is the LESA spectrometer, equipped only with an He-Ne laser (633 nm). Therefore, the area of an application was limited only to selected sensitizers. We have modified the system by using many lasers in the UV–VIS range, which allows us to excite the fluorescence of virtually all chemical compounds, including those of biological origin.

The system locally determines in vivo the level of photosensitizer accumulation in any patient’s organs and tissues accessible for a fiber optic probe. The system is used during photodynamic therapy of intracavity, interstitial and superficial tumors, and for measurements of biological tissues’ autofluorescence.

The LIF method is indispensable for the analysis of photosensitizers that are constantly growing in the market. These include porphyrin derivatives, phthalocyanines, bacteriochlorines. They differ significantly in parameters—dose, time of irradiation commencement and level of accumulation. A properly conducted PDT method requires the control of parameters and conditions.

The LIF method has great potential in pulmonology. Preneoplastic changes (dysplasias) and early neoplastic stages (intraepithelial carcinoma—CIS, microinvasion) are difficult to detect using traditional bronchoscopic methods, as the lesions cover an area up to several millimeters in diameter and several cell layers (0.2–1 mm thick) [33].

In gynecology, photo diagnosis helps to precisely determine the location of precancerous lesions and malignancies of the vulva (e.g., vulvar lichen sclerosis), vagina and cervix. PDD enables the detection of hyperplastic at its early stages (Figure 12).

The autofluorescence method has a good chance of being successful in the diagnosis of various infections, skin pigmentation changes and metabolic disorders. Thus far, the widely used Wood’s lamp for observing changes in skin fluorescence is an important tool in dermatology. Currently, changes in fluorescence are determined only visually, which, combined with too low power density of the mercury lamps used, is a big limitation of the method. Some fungal infections caused by pathogenic fungi can be precisely diagnosed by fluorescence methods. The fluorescence spectra depend on the type of disease. The current level of diagnostics allows us to only link the characteristic color of luminescence with the type of infection.

In the case of an infection of the skin with the *Malassezia furfur* fungus, which causes tinea versicolor, the luminescence has a copper-orange color; the light of coral-red emission is characteristic of Erythrasma.

Real-time autofluorescence testing methods cover an increasing range of medical applications. Different fluorescence imaging systems in bronchoscopy (e.g., LIFE, Vancouver, Canada) [34] and dermatology have already been built. A good example is the use of a VELscope (Vancouver, Canada) [35] lamp to evaluate the pathological changes in the mucous membrane. Observation of the changes in the metabolism of the surface layers of the tissues lining the mouth is important because they come into direct contact with many carcinogens and are the starting point of oral cancer. The most common disorders in the oral cavity include leukoplakia, erythroplakia, lichen planus and submucosal fibrosis. The risk of neoplastic metaplasia for these lesions varies, but early detection and prompt treatment can prevent cancer development.

## Figures and Tables

**Figure 1 sensors-22-02956-f001:**
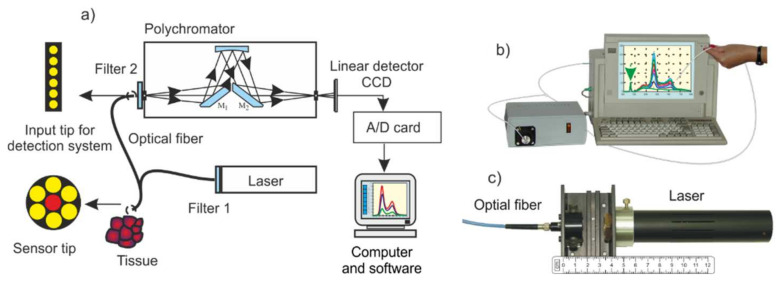
Fiberoptic fluorescence analyzer: (**a**) optical scheme, (**b**) view, (**c**) laser with input optics.

**Figure 2 sensors-22-02956-f002:**
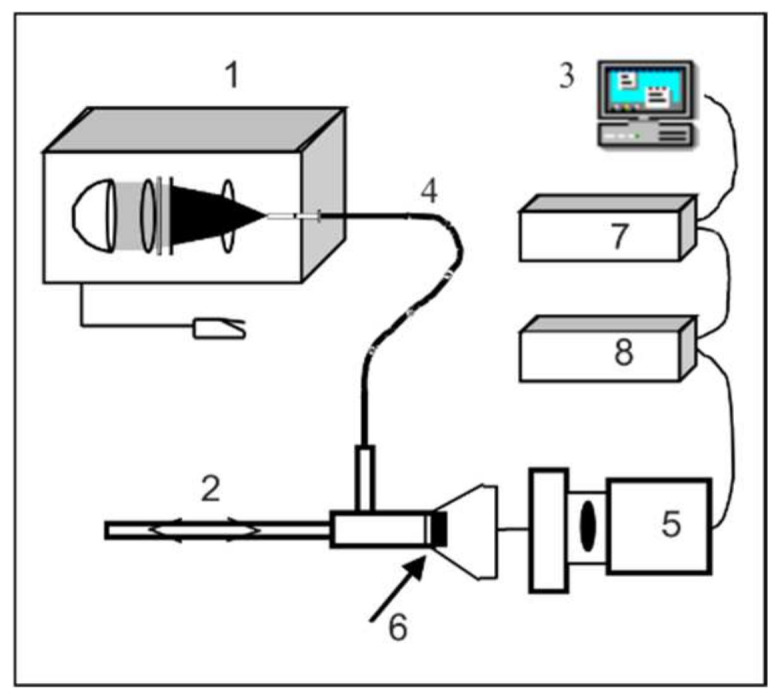
Optical diagram of the fluorescence imaging system: 1—light source with a violet filter (λ = 405 ± 25 nm), 2—endoscope, 3—monitor, 4—liquid optical fiber, 5—CCD camera, 6—optical filter, 7—computer, 8—video.

**Figure 3 sensors-22-02956-f003:**
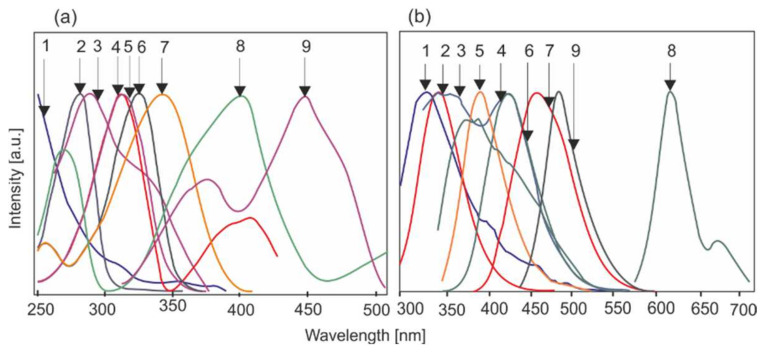
Spectral characteristics of potential endogenous fluorophores: 1—collagen, 2—tryptophan, 3—elastin, 4—pyridoxamine phosphate, 5—pyridoxine, 6—pyridoxal phosphate, 7—NADH, 8—protoporphyrin IX, 9—FAD. (**a**)-absorpton spectra, (**b**) emission spectra.

**Figure 4 sensors-22-02956-f004:**
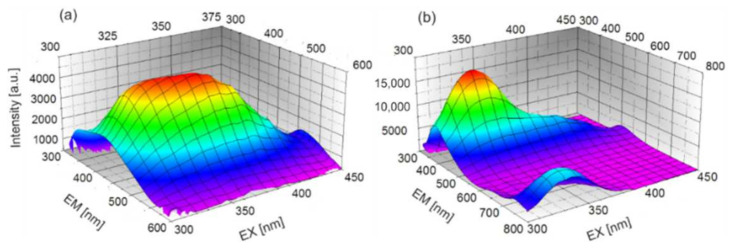
Emission–excitation characteristics (EM–EX) of (**a**) enamel and (**b**) dentin.

**Figure 5 sensors-22-02956-f005:**
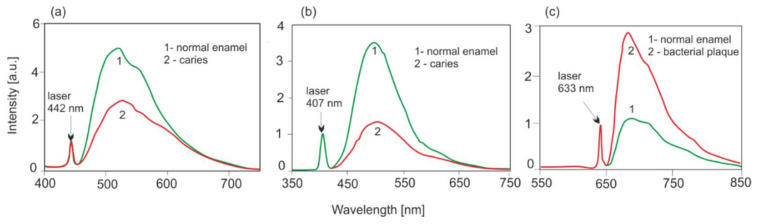
Influence of the excitation wavelength with laser radiation on changes in the level of enamel fluorescence: (**a**) laser 442 nm, (**b**) laser 407 nm, (**c**) spectra of bacterial plaque with excitation 633 nm.

**Figure 6 sensors-22-02956-f006:**
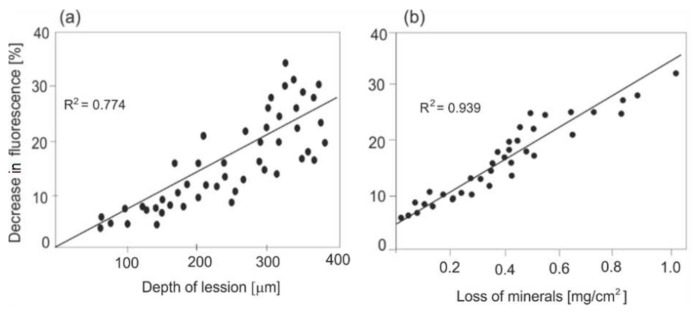
Influence of the (**a**) depth of the lesion and (**b**) loss of minerals on the decrease in fluorescence intensity.

**Figure 7 sensors-22-02956-f007:**
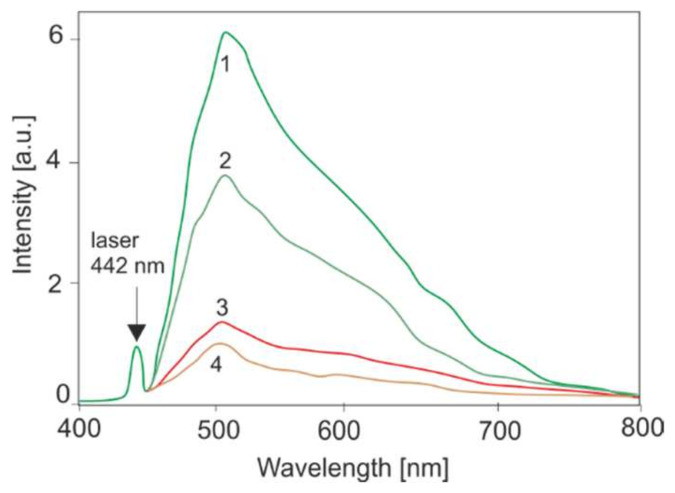
Decrease in autofluorescence in the mesothelium (1,2—normal tissue, 3,4—tumor).

**Figure 8 sensors-22-02956-f008:**
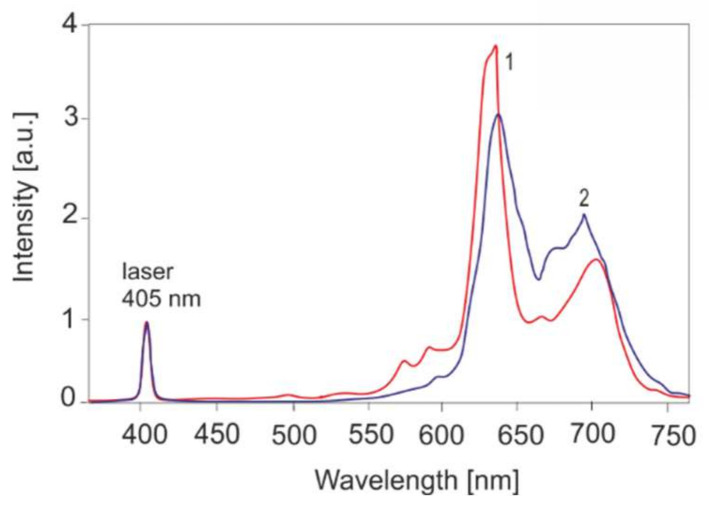
Fluorescence of metalloporphyrins: enamel plaque (1) and skin in senilis keratosis (2).

**Figure 9 sensors-22-02956-f009:**
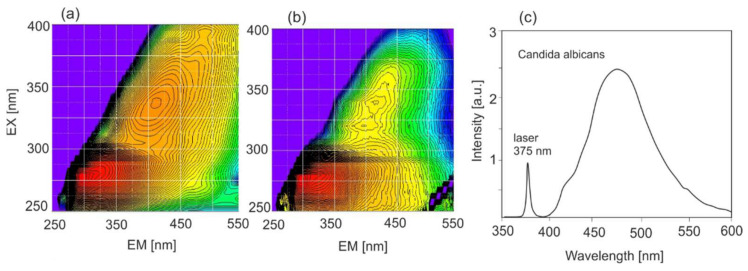
Spectral characteristics of selected fungi: (**a**) EM–EX map of *Penicyllium chrysogenium*, (**b**) EM–EX map of *Aspergillus flavus*, (**c**) LIF spectrum of *Candida albicans*.

**Figure 10 sensors-22-02956-f010:**
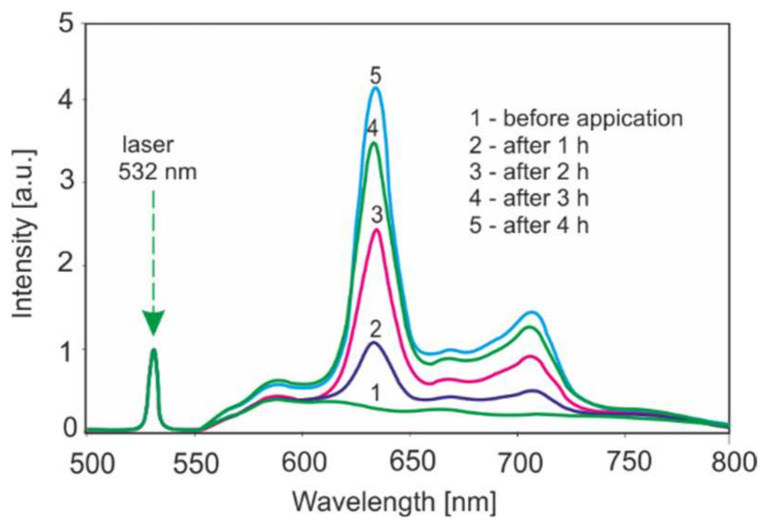
Kinetics of PPIX accumulation in alteration of skin actinic keratosis.

**Figure 11 sensors-22-02956-f011:**
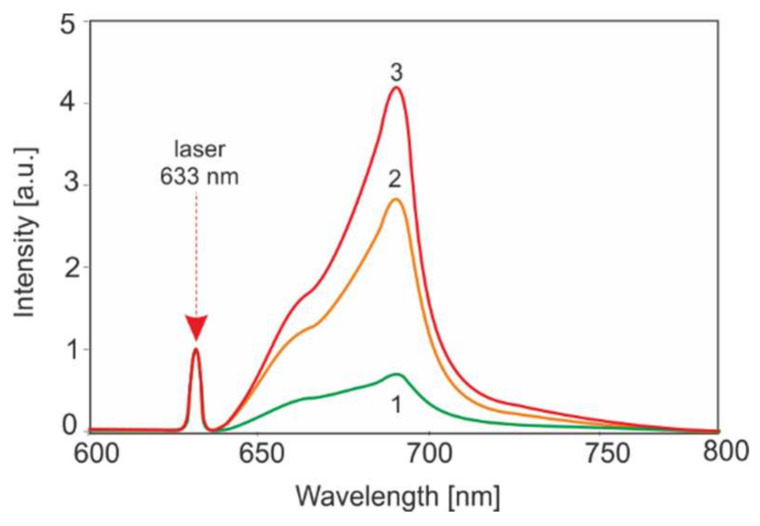
Fluorescence spectra of Merkel tumor with introduced PP(Ala)_2_(Arg)_2_: 1—healthy tissue, 2—tumor on the periphery, 3—tumor in the center of lesions.

**Figure 12 sensors-22-02956-f012:**
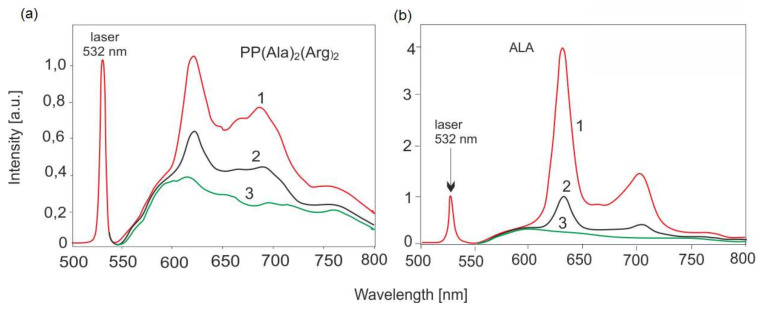
Comparison of accumulated porphyrin concentrations in (**a**) cervical cancer PP (Ala)_2_(Arg)_2_ and (**b**) vaginal (ALA): 1—tumor in the center of lesions, 2—tumor on the border, 3—normal tissue.

**Figure 13 sensors-22-02956-f013:**
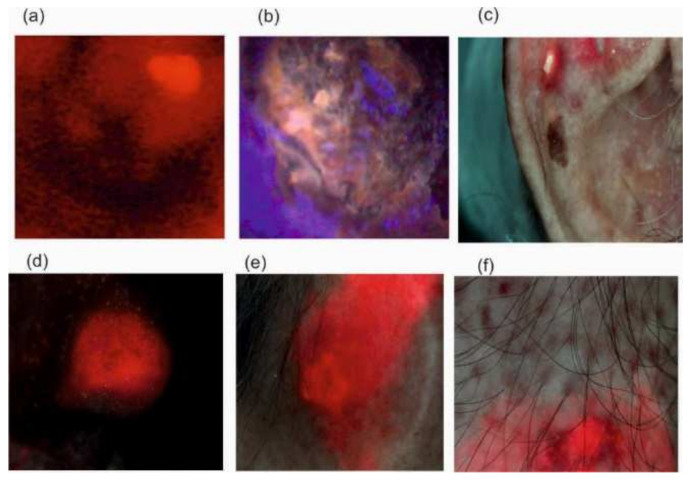
Fluorescent images of (**a**) vaginal, (**b**) cervical, (**c**) basal cell carcinoma of head, (**d**) squamous cell carcinoma of nose, (**e**) actinic keratosis of skin, (**f**) after ALA applications.

**Figure 14 sensors-22-02956-f014:**
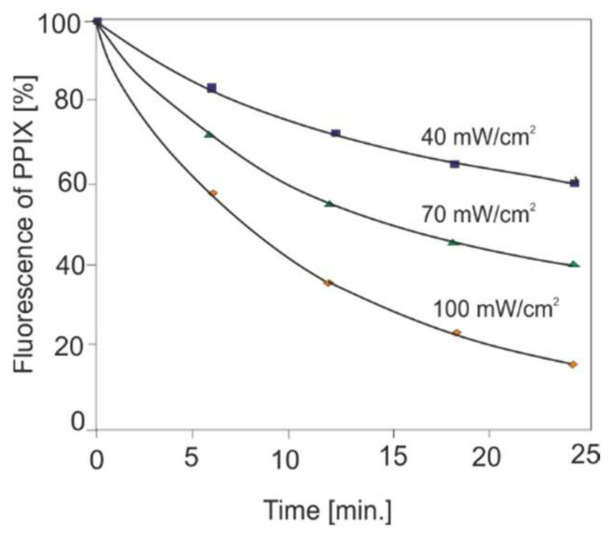
Photobleaching effect during the irradiation of skin actinic keratosis with PPIX.

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
