# Peer review of "Applications of Laser-Induced Fluorescence in Medicine"

_sensors, 2022, doi:10.3390/s22082956_

Round 1

Reviewer 1 Report

This manuscript titled “Applications of laser-induced fluorescence in medicine” describes advantages and examples of applications of laser-induced fluorescence method. The authors show the difference of fluorescence spectra of several samples for dental diagnostics and PDT and so on. 

The usefulness of the method employed is basically sound. The paper provides examples of LIF applications as review like but the paper does not contain significant finding in this study itself. I think the manuscript is informative but scientific novelty is limited. 

I have the following concerns.

  1. The title is very review like. if the authors intended to present as review, the paper should be submitted as a review article, not an original research article.
  2. In several figures, the authors compare two spectra of LIF measurements. The quantitative aspect such as numerical comparison with deviation/error or some statistical evaluation is needed.
  3. There is no scale bars in Figure13.
  4. In Figure 6(a), unit on X-axis is “mm” but it seems to be micro-meter.

Author Response

Dear Sir/Madame

Thanks a lot for your review. Let me start with detailed comments.

In Figure 6(a), unit on X-axis is “mm” but it seems to be micro-meter.

There are no scale bars in Figure13.

Obviously, in Figure 6, the thickness dimension should be in mm, not mm. As for Figure 13, I understand that the remark concerns the dimensions of the analyzed surfaces. Please note that the imaging is for objects as different as the head and the enodothelim, hence the magnifications are different. Actual sizes are not given in fluorescent images. We added a statistical evaluation to the studies in which we compared the level of fluorescence between healthy and diseased tissues (Fig. 7). In other cases (Figures 10, 11, 12), such an assessment does not make sense, because the level of fluorescence of photosensitizers depends on many factors: time after application, concentration of the sensitizer, and differences between individual patients. The quantitative aspect is also presented in Figure 6. The remaining figures are only examples of the LIF technique used. We also expanded the literature citation.

In several figures, the authors compare two spectra of LIF measurements. The quantitative aspect such as numerical comparison with deviation/error or some statistical evaluation is needed.

Manuscript was supplied with statistics. Lines 204-207.

The title is very review like. if the authors intended to present as review, the paper should be submitted as a review article, not an original research article.

The article was written with a topic relevant to the Special Issue "Biomedical Sensors for Functional Mapping: Techniques, Methods, Experimental and Medical Applications". The article presents the construction of fluorescent sensors for real-time analysis of various biological tissues. Optical fibers are adapted to cooperate with endoscopes, hence it is possible to analyze all human organs. This article presents the original results and is not a review article

Moreover, it is difficult to find in the literature the results of research carried out with analogous techniques or methods of analysis. All results come from our own research. We have presented all the potential uses of fluorescent sensors and not scattered examples. The constructed laser, fiber optic fluorescence analyzers have been used in many Polish dermatology, gynecology and dental clinics. These are compact systems and not laboratory stands. I am enclosing a view of the LIF analyzer with excitation of the He-Ne (633 nm) laser and a touch screen.

Please note that there is only a He-Ne laser in the original LESA, laboratory spectrometer. We have modified the system by using many lasers in the UV-VIS field, which allows us to test caries, sensitizers in the PDT method or even fungi in dermatology

Hundreds or even thousands of articles have been devoted to fluorescence methods in medicine, and it is impossible to write an actual review article. I absolutely do not agree that the article lacks scientific news. Please indicate me any article which, for example, presents the full spectral characteristics of enamel and dentin in the form of EX-EM matrices or the correlation between the quantitative change in enamel (Fig. 6) changed by caries and the decrease in the level of fluorescence was investigated. These are very time-consuming tests that require appropriate equipment. Another example is the study of the photochemical decomposition of PPIX under the influence of various light intensities in the PDT method (Fig. 12) or the quantitative on-line studies of photosensitizers and endogenous fluorophores. (Fig. 8,9) Without testing the level of sensitizers in the PDT method, it is not possible to carry out the correct therapy and the treatment procedure has to be repeated many times.

There are already several expensive and complicated fluorescence imaging systems for medical applications (e.g. bronchoscopy) on the market. However, their price ranges from EUR 100,000 to EUR 300,000, which is a prohibitive price for many. Our goal was to present how the necessary research can be carried out efficiently and cheaply.

While I agree that the topic suggests some kind of revision of the LIF method, it has been difficult to simply capture a combination of research from several areas. I would be grateful for a possible suggestion to modify the topic to fully correspond to the content of the article.

Yours sincerely

Mirosław Kwaśny

Reviewer 2 Report

The paper entitled: Applications of laser-induced fluorescence in medicine, by: M. Kwaśny and A. Bombalska, presents selected technology developed and used by the authors in some medical applications. The paper is interesting as an article of opinion, more than a classical article of research about a focused type of problem. Instead, the paper presents some applications of laser induced fluorescence in dispersed medical problems. I think that the paper should be more intended as a short review about the techniques, in that case the number of references is extremely short, and that should be improved. Along the article, the authors give information about the importance of diverse types of applications with references scattered every few paragraphs, so many of them are not referenced. The paper should be completed with references to constitute a short revision article or focus on a particular aspect of research with conclusions about the improvement of knowledge about the aspect.

Author Response

Dear Sir/Madame

Thank you very much for your review. I would also like to share my own comment. The article was written with the subject matter relevant to the Special Issue "Biomedical Sensors for Functional Mapping: Techniques, Methods, Experimental and Medical Applications" in mind. The article presents the construction of fluorescent sensors for the analysis of various biological diseased tissues in real time, without collecting biological material, and all possible applications of the LIF technique in medical applications.

The aim was to present my own original results and not to write a review article. The specific problem was to solve the quantitative real-time (on-line) measurement of fluorophores in any human organ. We have presented all the potential uses of fluorescent sensors, not just scattered examples. The constructed laser, fiber optic fluorescence analyzers have been used in many Polish dermatology, gynecology and dental clinics. These are compact systems and not laboratory stands. For an example I am showing another compact LIF circuit with He-Ne laser excitation (EX = 633nm).

In vivo evaluation of the level of fluorophores in real time, we can distinguish only the technique of imaging and the study of fluorescence spectra. But only LIF spectroscopy allows for accurate quantitative analysis. There are so many existing techniques. We did not develop the imaging technique, these systems are rather difficult to classify as classic sensors. There are literally three imaging systems for bronchoscopy and a few simple imaging systems for dentistry and dermatology on the market without quantification. Our goal was to develop a technique for quantifying fluorophores in all possible medical cases - for caries testing, PDT sensitizers, and cancer by autofluorescence (endogenous sensitizers). The only such commercially available system is the LESA spectrometer, equipped only with a He-Ne laser. We have modified the system by using many lasers in the UV-VIS field, which allows us to test caries, PDT allergy sufferers or even fungi in dermatology.

However, please note how important and complex the research we have carried out is

  • Correlation between the level of fluorescence of healthy enamel changed by the early stages of caries and the loss of minerals or the depth of the lesion
  • Studies on the kinetics of accumulation, excretion and photochemical decomposition of photosensitizers in the PDT method. Such data are the basis for the treatment with this method and prevent the need to repeat the process many times
  • Examination of the level of endogenous fluorophores in neoplastic tissues or other lesions
  • We indicated the possibilities of detecting fungi in dermatology.

As suggested, we have expanded the number of works cited in recent years.

Your sincerely

Mirosław Kwaśny

Round 2

Reviewer 1 Report

Thank you for your response. I agree with that the article contains original data and the data itself is scientifically new results. However, I think that the statement "what is figured out from your experimental results" was not clear throughout the manuscript. For example, (1) if you developed new measurement equipment by replacing excitation laser, as written in your letter,I think this issue should be mentioned in the manuscript preferably with the comparison against current system. (2) If the authors want to claim the usefulness of the newly developed application concept of LIF (e.g. dental diagnostics), I believe that there should be some discussion regarding the advantage of the LIF revealed by this study, based on the original data taken. 

Author Response

Dear Sir/Madame,

Thank you very much for the next review and very valuable comments with which I completely agree. Now, when I calmly analyze the content of this article, I come to the conclusion that it was better to limit myself to one issue, e.g. PDT or diagnosis of caries. This way the content would be more consistent. I have included additional sentences in the manuscript based on your comments. (lines 307-311, 325-331).

I would also like to refer to the assessment of the style in English. Before submitting this article, we paid a publishing house that professionally corrected style and grammar. We understand that this style can never be as high as the native language.

Your sincerely                                           

Mirosław Kwaśny

Reviewer 2 Report

The paper entitled: Applications of laser-induced fluorescence in medicine, by: M. Kwaśny and A. Bombalska, presents selected technology developed and used by the authors in some medical applications. The paper is interesting as an article of opinion, more than a classical article of research about a focused type of problem. The paper presents some applications of laser induced fluorescence in dispersed medical problems. Along the article, the authors give information about the importance of diverse types of applications with selected references about them. The paper has now been completed with sufficient references to constitute a hybrid short revision/accounts article focused on a particular aspect of research with conclusions about the improvement of knowledge about every particular aspect and so it can be published as it is.

Author Response

Dear Sir/Madame

Thank you for reviewing our manuscript. It is a privilege to publish in Sensors. Thank you for valuable and kind revision.

Sincerely yours

Mirosław Kwaśny
